# TDP-43 Proteinopathy Specific Biomarker Development

**DOI:** 10.3390/cells12040597

**Published:** 2023-02-12

**Authors:** Isabell Cordts, Annika Wachinger, Carlo Scialo, Paul Lingor, Magdalini Polymenidou, Emanuele Buratti, Emily Feneberg

**Affiliations:** 1Department of Neurology, Klinikum Rechts der Isar, School of Medicine, Technical University of Munich, 81675 Munich, Germany; 2Department of Quantitative Biomedicine, University of Zurich, 8057 Zurich, Switzerland; 3Deutsches Zentrum für Neurodegenerative Erkrankungen, DZNE, 81377 Munich, Germany; 4Munich Cluster for Systems Neurology (SyNergy), 81377 Munich, Germany; 5Molecular Pathology, International Centre for Genetic Engineering and Biotechnology, 34149 Trieste, Italy

**Keywords:** TDP-43, biomarker, neurodegeneration, amyotrophic lateral sclerosis, frontotemporal dementia, dementia, cerebrospinal fluid, biofluid

## Abstract

TDP-43 is the primary or secondary pathological hallmark of neurodegenerative diseases, such as amyotrophic lateral sclerosis, half of frontotemporal dementia cases, and limbic age-related TDP-43 encephalopathy, which clinically resembles Alzheimer’s dementia. In such diseases, a biomarker that can detect TDP-43 proteinopathy in life would help to stratify patients according to their definite diagnosis of pathology, rather than in clinical subgroups of uncertain pathology. For therapies developed to target pathological proteins that cause the disease a biomarker to detect and track the underlying pathology would greatly enhance such undertakings. This article reviews the latest developments and outlooks of deriving TDP-43-specific biomarkers from the pathophysiological processes involved in the development of TDP-43 proteinopathy and studies using biosamples from clinical entities associated with TDP-43 pathology to investigate biomarker candidates.

## 1. Introduction

Amyotrophic lateral sclerosis (ALS) and frontotemporal dementia (FTD) are diagnosed based on clinical criteria. Often, this leads to a delay between the onset of symptoms and diagnosis up to one year in ALS and 5 years in FTD, narrowing the window where potentially disease-modifying therapies would likely be most effective [1,2,3,4,5]. In order to intervene early and effectively in those diseases, biomarkers reflective of the underlying core pathobiology are needed [6,7]. These biomarkers will serve to identify the right cohorts for therapeutic targets and to determine the efficacy of therapies and clinical trials more accurately [8].

Ubiquitinated cytoplasmic aggregates of the nuclear 43 kDa transactive-response, DNA-binding protein, TDP-43, is the core pathobiology in almost all cases (97%) of ALS [9]. TDP-43 pathology is also found in 50% of frontotemporal lobar degeneration (FTLD) cases, most commonly associated with the clinical syndrome of behavioral variant frontotemporal dementia (bvFTD), while the rest is associated with tau protein pathology [10,11,12]. The interest in TDP-43 proteinopathies has further broadened since TDP-43 neuropathology has been found in 30% of Alzheimer’s disease (AD) cases [13,14]. Neuropathological evidence of TDP-43 pathology in AD has recently been defined as limbic-predominant age-related TDP-43 encephalopathy (LATE), which can present itself with or without coexisting hippocampal sclerosis pathology [15].

The clinical heterogeneity ranging from pure motor (ALS) to cognitively impaired (ALS-FTD) or dementia (FTD, LATE) phenotypes that are associated with TDP-43 proteinopathy is striking and likely associated with a distinct topographical distribution of TDP-43 pathology in the brain [16,17]. Pathological TDP-43 in ALS predominantly involves the spinal cord and the motor cortex but can be more widespread in the frontal and temporal lobe structures [18]. In FTD, TDP-43 is mostly found in the orbitofrontal cortex, but can exceed into the frontal and temporal cortices before involving the motor system, visual cortex, and cerebellum [19]. In contrast, TDP-43 pathology in LATE is mostly localized to the hippocampus and amygdala, suggesting that a selective cellular and regional vulnerability contributes to the disease [15]. There is, indeed, increasing evidence that TDP-43 functionality can be shaped depending on cell type [20,21]. Different types of TDP-43 pathology can be found in ALS/FTD clinical phenotypes; for example, type B pathology with large cytoplasmic inclusions and few dendritic neurites is mostly associated with ALS and bvFTD [22]. The clinical heterogeneity among FTD (bvFTD or primary progressive aphasia (PPA)) and the clinical similarity of LATE to AD makes it difficult to accurately predict the underlying pathology in life. At present, stratifying patients by core pathology and their assignment in treatment trials designed to interfere with TDP-43 pathobiology is not yet possible, except for rare well-defined genetic cases [23]. Hence, it is paramount to develop TDP-43 molecule-specific biomarkers.

Here, we will review the latest developments and outlooks of deriving TDP-43-specific biomarkers from: (1) pathophysiological processes involved in the development of TDP-43 proteinopathy, and (2) biofluid studies using biosamples from clinical entities associated with TDP-43 pathology. We identified the existing evidence through author knowledge and PubMed searches from the database’s inception to April 2022.

## 2. TDP-43 Pathobiology Informed Biomarker Development

### 2.1. Cellular Homeostasis and Aggregation Propensity

TDP-43 belongs to a family of heterogeneous nuclear ribonucleoproteins (hnRNP) [24]. Structurally, TDP-43 consists of a structured N-terminal domain (NTD) involved in physiological self-oligomerization, followed by two tandem RNA-recognition motifs (RRM), which bind to certain nuclear transcripts and therefore regulate important DNA/RNA metabolism functions (Figure 1a) [25,26,27,28,29]. The RRMs are flanked by N- and C-terminal regions. The C-terminal end of TDP-43 is a conserved, glycine-rich region important for the function of alternative splicing, protein–protein interactions and to promote the formation of phase-separated RNA granules [30,31,32,33,34]. The C-terminal domain of TDP-43 is also affected by most ALS-associated genetic variants [35,36], which, together with changes in RNA levels, can enhance the formation of dynamic liquid-like condensates [37,38]. In addition, an intrinsically disordered (IDR) prion-like region within the C-terminus harbors a tendency to self-assemble (“sticky ends”) and to form different oligomeric states [39,40]. There is, therefore, a critical equilibrium between TDP-43′s physiological function and its soluble form [25]. TDP-43 preferentially binds to long UG-rich sequences, including its own mRNA, via N-terminal homo-dimerization and its C-terminal conserved region [28,41,42,43]. Many of these regions have been found to be selectively modified by a variety of post-translational modifications, which include phosphorylation, acetylation, and SUMOylation [44]. Many of these events are now known to act as drivers of the pathology mostly by modifying the phase separation properties of this protein [45,46].

Pathological processes that disturb the selective binding mechanism of TDP-43 enhance homonymous binding and self-aggregation, disturb its autoregulation, and lead to increased concentrations of TDP-43-aggravating protein accumulation (Figure 1b) [47,48]. Its potential for dimerization and oligomerization and the transcellular transmission of such TDP-43 oligomers has been shown in in vitro experiments [49,50,51]. Such self-aggregation and seeding propensities of misfolded proteins have been exploited for diagnostic purposes in prion disease, and later, a-synuclein and ß-amyloid pathologies using real-time quaking-induced conversion (RT-QuIC) [52,53,54,55]. Therefore, RT-QuIC for inducing TDP-43′s intrinsic aggregation propensity into non-amyloid, fibrillar aggregates, seemed a plausible technique for the detection of pathological TDP-43 species in patient biosamples [56,57,58]. A major requisite for developing RT-QuIC was achieved by producing functionally intact recombinant full-length and C-terminal fragmented TDP-43 species, which after mechanical and heat activation were able to seed amorphous TDP-43 aggregates or fibrillary structures in a measurable time frame [59]. This technique was then employed for cerebrospinal fluid (CSF) samples from patients with genetically confirmed TDP-43 proteinopathy and normal controls using the recombinant C-terminal fragment. A positive RT-QuIC reaction was obtained from TDP-43 positive samples with a high sensitivity of 94% and specificity of 85%. However, the limitations of this study were the false positive results, possibly originating from the presence of low levels of TDP-43 aggregates in healthy subjects, or by the heterogeneous presence of contaminants in the CSF, which needs to be tested in future studies. Another technique that uses aggregate formations of TDP-43 as a readout for TDP-43 proteinopathy is immuno-infrared sensor technology [60]. Here, TDP-43 is captured from CSF using oligoclonal antibodies targeting the aminoacid epitopes 130–275 of TDP-43. Bound TDP-43 is then measured by an infrared beam to assess the absorbance maximum indicative of the presence of an alpha-helical or β-sheet structure. The comparison of CSF from patients with ALS and Parkinson’s disease (PD), as well as neurological controls, demonstrated an increase in the β-sheet structures in CSF samples of ALS compared to PD and control samples. This approach displayed a diagnostic accuracy to discriminate ALS from PD (or controls) with 89% (89%) sensitivity and 77% (83%) specificity, with the caveat that this study lacked direct evidence of antibody-bound TDP-43 by immunoblotting. In summary, the self-aggregation propensity of TDP-43 has become a valuable tool for developing in vivo diagnostics; whether these techniques have the potential to be standardized for routine measurements requires further research.

### 2.2. Post Mortem TDP-43 Proteinopathy

The pathology of TDP-43 in ALS and FTD is characterized by its mislocalization from the nucleus to the cytoplasm of neurons and glia cells, where it is found post-translationally modified by phosphorylation and ubiquitination, as well as N-terminally cleaved into smaller C-terminal fragments (CTF) [9]. Neuropathological studies in ALS and FTD have shown that the phosphorylated form of TDP-43 is distributed in a relatively stereotyped spectrum of anatomical patterns, which has led to the definition of neuropathological stages [18,19]. In addition, the subtypes of characteristic TDP-43 staining were classified as different types (A–E) of TDP-43 proteinopathy, often characteristic for certain clinical syndromes within the ALS-FTD spectrum [61,62]. Both the regional localization of TDP-43 and the type of TDP-43 pathology are associated with a variable degree of cellular degeneration reaching from moderate whole-brain atrophy in ALS [63] to more severe focal frontal/temporal tissue loss in FTD [64].

Ultimately, a major breakthrough in neuropathology was achieved when the serial fractionation of post mortem brain tissue led to the extraction of insoluble proteins, which specifically recovered pathological TDP-43 from the affected brain regions [9]. A characteristic biochemical pattern of TDP-43 from the insoluble protein fraction includes a high molecular weight smear, which likely reflects phosphorylated and ubiquitinated TDP-43, phosphorylated full-length TDP-43 with a molecular mass size of 45–50 kDa and 60 kDa, and truncated forms at 24–26 kDa, identified as C-terminal fragments of TDP-43. C-terminal fragments and post-translational modifications of TDP-43 have been a prominent hypothesis proposed for TDP-43-associated neuronal toxicity and pathobiology [65,66]. Several studies therefore exploited the enrichment of pathological TDP-43 species from post mortem tissue in order to find a pathology-specific TDP-43 biomarker [9,10,67]. By mass spectrometry analysis, TDP-43 post-translational modifications or pathological peptide sequences were identified without the limitations of antibody-based methods. This approach enabled the identification of the pathological phosphorylation sites of TDP-43 predominantly located at its C-terminal glycine-rich domain [67,68]. These findings confirmed the disease-specificity of phosphorylated TDP-43 from previous studies using phosphorylation dependent TDP-43 antibodies [69]. Regarding phosphorylation, a recent important development has also occurred following the isolation of TDP-43 fibrils from post mortem brain samples and their analysis using mass-spec. This approach has allowed us to start investigating possible PTM differences between the different types of TDP-43 pathology in GRN vs. C9ORF72 cases [70]. Further, a proteomic approach using post mortem tissue from different subtypes of TDP-43 proteinopathies allowed the identification of subtype-specific insoluble proteomes [71], while a similar approach using ALS post mortem tissue enabled the characterization of peptide sequences that were representative of TDP-43 protein fragments [67]. This work showed that peptides with a non-enzymatic N-terminal truncation site are representative of an endogenously cleaved C-terminal fragment, while less common peptides with a C-terminal truncation site represented N-terminal TDP-43 fragments. This was then taken forward to develop methods by targeted mass spectrometry for the absolute quantification of such fragment peptides [72,73,74]. Heavy isotope-labelled peptides of the reference peptides were used to absolutely quantify endogenous TDP-43 peptide levels. Using this method, it was shown that in the motor cortex of ALS patients with TDP-43 proteinopathy, the C:N-terminal peptide ratio of TDP-43 was significantly increased compared to both normal and other neurodegenerative disease control groups, suggesting a measurement for the enrichment of pathological C-terminal TDP-43 fragments in ALS [72]. The pathological cleavage of TDP-43 was further confirmed by measuring the N-terminal truncation site-specific TDP-43 peptides. This included the identification of a novel N-terminal truncation of a C-terminal fragment. Of note, the cleavage of TDP-43 was found in ALS, but also in the motor cortices of AD, that were later confirmed with LATE neuropathological changes [72]. In another work, multiple peptides spanning TDP-43 on structural and pathological relevant sites, including the N-terminus, both RNA-binding domains (residues 101–262) and the domains of TDP 35 kDa, and TDP 25 kDa fragments (residues 90–414 and 220–414) were developed [73]. Together with an enrichment process of functional RNA-binding TDP-43 using aptamers prior to mass spectrometric analysis, the sensitivity of detection was increased, and this enabled the improved quantification of the cleavage patterns of TDP-43 [74]. In conclusion, shifting the detection of TDP-43 on a proteotypic peptide level enables to form the basis for the absolute quantification of N-terminal and C-terminal TDP-43 fragments in future proteomic studies of complex matrices, such as blood and CSF, as well as the development of novel antibodies recognizing the pathological forms of TDP-43.

### 2.3. TDP-43 Loss of Function Mechanisms

While cytoplasmic aggregation with a toxic gain of function is a proposed mechanism of TDP-43 pathobiology, a second mechanism of toxicity is the nuclear loss of TDP-43 leading to altered RNA/DNA metabolism [75,76]. It is also suggested that the nuclear loss of TDP-43 could be an early event in the development of the disease preceding cytoplasmic TDP-43 aggregate formation [77,78]. TDP-43 acts as a splicing repressor of non-conserved intronic sequences, which, when disturbed, results in the inclusion of normally cryptic exons into mature RNAs (Figure 1b) [36]. This can lead to premature stop codons, premature polyadenylation, or the degradation of certain transcripts [79]. In line with this loss of function mechanism, a major alteration of the RNA levels of stathmin 2 (STMN2) and unc-13 homolog A (UNC13A) has been recently described [80,81]. Truncated and non-functional STMN2 mRNA, caused by premature polyadenylation resulting in an eightfold reduction of normal STMN2 levels, was found in the TDP-43 proteinopathy positive postmortem brain regions of FTD patients [82]. STMN2 is important for the stabilization of long axons in neurons and the loss of STMN2 has been associated with toxicity in ALS [83,84]. A similar pathology has been described for UNC13A, where TDP-43 depletion in neuronal cell lines lead to reduction of UNC13A transcripts through nonsense-mediated decay [81]. The repression of cryptic exon inclusion in UNC13A was present in neuronal nuclei with TDP-43 proteinopathy in post mortem brains from patients with ALS and FTD and the reduction of UNC13A protein after TDP-43 depletion was shown [85]. UNC13A is a synaptic protein important for glutamatergic neuronal transmission [86,87,88]. Pathological UNC13A polymorphisms in the same intronic region affected by mis-splicing have been associated with an increased risk in developing ALS/FTD, as well as a shorter survival [89,90,91,92]. In neurons overexpressing TDP-43 and post mortem brains with TDP-43 proteinopathy, the loss of nuclear TDP-43 was specifically associated with the upregulation of the synaptic neuronal pentraxin 2 (NPTX2) mRNA [93]. TDP-43 regulates the levels of NPTX2 mRNA through binding at its GU-rich region [94]. This function was shown to be disrupted in FTLD brain tissue where NPTX2 was also found aberrantly accumulated in neurons [93]. From a biomarker point of view, it is important to note that NPTX2 has also been found decreased in CSF samples of symptomatic FTD mutation carriers [95]. While most patients had TDP-43-associated chromosome 9 open reading frame 72 (C9ORF72) gene and granulin (GRN) gene mutations, NPTX2 was also lower in microtubule-associated protein Tau (MAPT) gene mutation carriers. Therefore, the specificity of NPTX2 as a biomarker for TDP-43 pathology is unclear and requires further investigation.

A further example of using a transcriptome-driven approach to develop non-invasive prognostic biomarkers for disease has been demonstrated for ALS [96]. In this approach, gene expression was derived from laser captured spinal motor neurons with TDP-43 pathology, and the data was bioinformatically processed into the modules associated with disease progression and upstream genetics. Measuring TDP-43 proteinopathy-dependent transcript alterations could therefore be a promising biomarker for discerning TDP-43 proteinopathies from other pathologies in life. In addition to the quantification of full-length protein levels, peptidomics is a promising tool to characterize protein isoforms and truncation signals that may arise from transcript alterations by including the search for unspecifically cleaved peptides. Data from proteomic studies of CSF samples so far have not been successful to detect STMN2 in either ALS patients or non-neurodegenerative diseases [97,98]. Alternative detection methods, such as hypothesis-driven targeted proteomics may therefore be considered in the future to enhance the sensitivity to detect a protein of interest by absolute quantification even in more complex matrices such as CSF [99].

### 2.4. TDP-43 Proteinopathy Induced Mitochondrial Dysfunction and Inflammatory Response

It is known that TDP-43 specifically interacts with mitochondrial proteins and alters mitochondrial dynamics [100]. Pathological TDP-43 fragments have also been found in mitochondrial protein fractions. However, these changes might be decreasing once neurodegeneration is severe, as recently shown in atrophic FTLD brain regions [101]. Nonetheless, many alterations in axonal mitochondrial transport and respiration have also been demonstrated in genetic models of ALS [102,103]. Most interestingly, in a cellular model of ALS cytoplasmic mislocalization of TDP-43 itself triggers its localization to mitochondria and leads to the release of mitochondrial DNA (Figure 1b) [104,105]. This process destabilizes the mitochondria by mitochondrial membrane permeabilization which eventually activates an inflammatory response via the cGAS/STING pathway. Other than the activation of the inflammatory cGAS/STING pathway, this leads directly to the activation of the TANK-binding kinase 1 (TBK1). In support, the TBK1 loss-of-function mutations have been linked to the development of both ALS and FTD TDP-43 proteinopathies [106]. While cGAS/STING activates TBK1, which then triggers a type 1 interferon response, mutations lead to haploinsufficiency of TBK1, meaning decreased TBK1 activity. Another interesting observation is that even loss of function of TBK1 seems to have an opposing effect in an ALS animal model [107]. The fact that TBK1 is directly associated with TDP-43 proteinopathy through the cGAS/STING pathway and can clinically lead to both diseases, ALS and FTD, and is eventually differentially regulated in early and late-stage disease, makes it an interesting pathway to follow for biomarker derivation. Inflammatory response, as well as evidence for inflammatory components that shape specific clinical and neuropathological patterns in the ALS-FTD-spectrum syndromes underpin the role of non-cell autonomous in ALS-FTD TDP-43 proteinopathy [108]. Among different inflammatory markers, chitinases, including chitotriosidase (CHIT1), chitinase-3-like-1 (CHI3L1), and chitinase-3-like-2 (CHI3L2), have been found to be significantly elevated in CSF of ALS patients compared to other neurodegenerative diseases and CHIT1-positive immunostaining was found in ALS spinal cord tissue, but not in the cortical tissue of ALS, FTD, or AD [109,110,111]. CHIT1 has also been found higher in the CSF of FTD patients with TDP-43 pathology [112]. However, the elevation of CSF CHIT1 levels in FTD are restricted to the clinical phase [113]. In addition, CHI3L1 in FTD correlates with frontal cognitive dysfunction [114,115].

Taken together the differential expression patterns of the chitinases in the ALS-FTD spectrum suggests that other factors, such as the severity of the disease, progression, or neurodegeneration pattern may account for differences rather than simply reflecting a reliable signal for TDP-43 proteinopathy.

### 2.5. Molecular Aspects of TDP-43 Pathology

TDP-43 pathology has been identified in a wide range of neurodegenerative diseases, both in cases inherited in a Mendelian pattern or sporadic patients, providing an exciting link between the pathophysiology of apparently sporadic and familial diseases. Generally, ~97% of all ALS and ~45% of FTLD cases involve TDP-43 aggregation [17,116]. In ALS, most sporadic patients and most patients carrying genetic variants, such as C9ORF72-, TARDBP-, OPTN-, UBQLN2-, and TBK1-ALS, show TDP-43 pathology [117,118]. Interestingly, in superoxide dismutase 1 (SOD1)-ALS, neuronal inclusions show immunopositivity for ubiquitin, but usually negativity for TDP-43. Similarly, in ALS due to genetic variants in the FUS RNA binding protein (FUS), a gene that is also encoding an RNA-processing protein involved in transcriptional regulation, TDP-43 positive inclusions are absent [119,120]. In FTD, TDP-43 pathology has been identified in Tau-negative sporadic cases (50%) as well as patients with genetic variants in, e.g., C9ORF72, GRN, and valosin-containing protein (VCP), but not in MAPT-FTD [17,117,118].

The discovery of key proteins might lead to the identification of candidate ALS-FTD genes, a prominent example being the identification of missense variants in TARDBP. Mutations in the gene encoding TDP-43 have, however, only being associated in less than 5% of ALS patients [35]. This is in line with the hypothesis that TDP-43 pathology may arise through multiple different mechanisms, emphasizing that ALS as a multistep disease, with fewer steps required in familial ALS [26,121]. On the other hand, the identification of mutated genes has a relevant impact on the development of novel disease biomarkers. For example, a loss of function mutation in cyclophilin A has been found in ALS and knocking out cyclophilin A in a mouse model led to the development of TDP-43 proteinopathy [122]. Cyclophilin A has also been found differentially expressed in ALS peripheral blood monocytes and plasma-derived extracellular vesicles [123,124]. These biomarkers could also be directly linked to the genetic defect, e.g., poly-GP in C9ORF72 expansion carriers or SOD1 protein concentrations in CSF of SOD1-ALS [125,126,127].

As alterations in RNA metabolism have emerged as one critical driver of ALS/FTD pathogenesis, the role of microRNA, small non-coding RNAs and key determinants of mRNA stability, has recently been studied [128,129]. Besides their important regulatory role in a variety of biological processes, miRNAs can also be released into the circulation by altered tissues as “fingerprints”, reflecting an activation of specific pathogenic pathways. Furthermore, they are easily detected in body fluids such as blood, CSF, or saliva, where they display remarkable stability. Interestingly, specific miRNA profiles have been identified in familial ALS, but also in pre-symptomatic mutation carriers [130]. At the level of a functional connection, it is known that the C-terminus of TDP-43 forms a complex with nuclear Drosha and cytoplasmic Dicer, which regulate miRNAs [131]. MiRNAs are gene expression modulators and influence cytoplasmic mRNA translation, indirectly giving evidence for a role of TDP-43 on cytoplasmic translation. Interestingly, TDP-43 has a direct effect on specific pri-miRNA processing and leads to differential expression of selected miRNAs [132,133], but the importance of these changes on disease onset and progression still remains to be largely evaluated.

In the future, however, additional types of non-coding RNAs could also be explored as ALS biomarkers. These include circRNAs that originate from back-splicing events during precursor mRNA processing and may act as miRNA sponges or to sequester RBP proteins, thus affecting protein and gene expression. Their usefulness as a biomarker mostly resides in the fact that circRNAs are very resistant to RNA exonucleases and are thus highly stable in cells. For example, a recent promising study of circRNAs in blood leukocytes of sporadic ALS patients has allowed us to identify four circRNAs whose expression may be differentially affected in patients compared to controls [134].

Finally, long noncoding RNAs (lncRNAs) could also represent another possible type of RNA that could be easily detected in blood and CSF of patients. As the name suggests, lncRNAs are long RNAs that are not normally translated but can affect gene expression by affecting epigenetic and transcriptional profiles of their target genes. To this date, lncRNAs have not received a very high level of attention in ALS, but differential expression of several lncRNAs has been reported to occur in peripheral blood monocytes of ALS patients [135]. It should also be noted that TDP-43 has been described to also affect the processing of several of these molecules, such as NEAT1 and MALAT, and to affect the severity of the disease in animal models [136,137].

## 3. Biofluid Studies of Clinical Entities Associated with TDP43 Pathology

The lack of studies aiming to develop TDP-43 disease-specific in vivo biofluid biomarkers by searching for common changes in TDP-43 pathology relevant patient groups is striking [138]. Especially in FTD, the clinical spectrum does not correlate with the distinct pathologies, except when the underlying pathology of FTD is predicted by the presence of an autosomal dominant mutation, which is found in only 20%–30% of the patients [139]. Here, therefore, we have wished to review biofluid studies that aim to identify biomarkers in patient groups that are clinically indicative, genetically, or autopsy-defined as TDP-43 proteinopathies using either applied unbiased proteomics or a data-driven approach.

### 3.1. Unbiased Shot-Gun Proteomics

A recent proteomic study compared CSF samples from FTD patients with TDP-43 or Tau proteinopathy, confirmed by autopsy or genetic testing to a group of patients with subjective memory complaint [138]. For the discovery experiments, only a small number of samples (*n* = 8 for Tau and *n* = 12 for TDP-43) were investigated. In total, ten biomarkers were observed to be differentially expressed between FTD-TDP and FTD-Tau, which were later validated. Specifically, Ubiquitin-like protein 3 was upregulated in FTD-TDP versus FTD-Tau, while α-Galactosidase A, Heat shock protein 8 and Kallikrein 7 were all downregulated in FTD-TDP compared to FTD-Tau. As a result, ten biomarkers were differentially regulated between the control group and FTD-TDP patients. However, only one candidate, catalase activity, was also found downregulated in FTD patients compared to the control group in the validation experiments, while other candidates failed. A second study investigated CSF samples from asymptomatic (*n* = 14), symptomatic (*n* = 14), ALS mutation carriers (SOD1, C9ORF72 and TARDBP), and sporadic ALS patients (*n* = 12) [97]. In these patient groups, changes were likely to reflect fast progressive neurodegeneration rather than TDP-43 pathology. However, a more specific ALS pathology was also discussed with regards to the observed upregulation of Ubiquitin C-terminal hydrolase-L1 (UCHL1) in patient’s CSF, which had been shown to be differentially expressed in ALS cases before [140,141,142,143]. Of particular interest, there was also the observation that several proteins involved in the splicing function were differentially regulated in ALS samples, which could indicate a dysfunction of TDP-43 [97].

### 3.2. Hypothesis Driven Investigation of Biomarkers

Specific biomarkers have been investigated across clinical cohorts of ALS-FTD spectrum disorders in comparison to other neurodegenerative diseases, such as AD and PD. Here we review biomarker studies in light of their potential to differentiate ALS, FTD-TDP, and eventually LATE, from FTD-Tau and AD patient cohorts (Table 1).

#### 3.2.1. Neurofilaments

A promising biomarker in neurodegenerative diseases is the light and phosphorylated heavy chain of neurofilaments (NfL and pNfH). Neurofilaments (Nf) are abundant cytoskeletal proteins of myelinated central and peripheral neurons [176]. Increased neurofilaments levels reflect axonal damage and are therefore unspecific. However, Nfs can aid to discriminate certain neurodegenerative disorders. In ALS, for example, NfL and pNFH are strongly elevated in CSF and serums with diagnostic sensitivities and specificities >80% [144,177,178]. Furthermore, Nfs were observed to be highly expressed in most FTD-spectrum diseases (including bvFTD, FTD-ALS, PPA, corticobasal syndrome (CBS), and progressive supranuclear palsy) with the highest values in FTD-ALS [146,147,148,149,167]. An autoptic or genetically secured FTD with TDP-43 pathology (FTD-TDP) showed higher Nf values when compared to FTD with Tau pathology (FTD-Tau) [150,179,180]. However, definite cut-off values to discriminate FTD-TDP from FTD-Tau have not been established and studies are restricted by the small numbers of pathology-confirmed cases. Further work will therefore be necessary to exactly evaluate the usefulness of Nf levels as a diagnostic disease biomarker in FTD pathology subtypes.

#### 3.2.2. Glial Markers: CHIT1, YKL-40, GFAP and TREM2

It is now widely accepted that neuroinflammation contributes to neurodegeneration in ALS and FTD, although it is unresolved whether the immune response causes, aggravates, or counteracts neurodegeneration [181]. Specifically, activated microglia and astrocytes produce several inflammatory proteins that have been explored in terms of biomarker value in neurodegenerative diseases. Among the first targets to be explored, Chitotriosidase 1 (CHIT1) was shown to be significantly increased in CSF of sporadic ALS patients [109,114]. While it is also increased in genetic ALS cases, CHIT1 is normal in genetic FTD cases and in asymptomatic gene carriers when compared to controls [113]. Another study showed higher CHIT1 levels in prion disease subtypes, FTD, and AD, but also higher values in FTD-TDP compared to FTD-Tau, but with the highest values in the ALS-FTD group [112]. Because CHIT1 is predominantly elevated in ALS, its levels are more likely to reflect the degree of motor neuron degeneration rather than reflecting the type of proteinopathy [109,113,114]. This conclusion has been supported by a comparison between FTD-TDP and a small group of “definite” FTD-Tau cases (definite PSP, CBD, and MAPT carriers) which revealed no differences. Similarly, levels of chitinase-3-like protein 1 (YKL-40) and glial fibrillary acidic protein (GFAP) were higher in prion disease subtypes, FTD, and AD compared to the controls [112]. In this study, it was also shown that the values of YKL-40 did not differ between FTD-TDP and FTD-Tau, but were increased when associated with ALS-FTD, while GFAP showed similar values among the clinical syndromes and proteinopathies. However, like p-Tau there is growing evidence that GFAP could be a biomarker for AD and serum GFAP levels could differentiate AD from bvFTD [182,183]. Another promising connection is represented by triggering receptor expressed on myeloid cells 2 (TREM2), which is a surface receptor expressed on brain microglia and whose genetic variants have been identified as risk factors for AD [184,185]. Since its description, this link has led to an extensive study of TREM2 biology and its potential role as an immune-related biomarker in neurodegenerative diseases. In CSF of AD cohorts, soluble TREM2 levels were reported to increase over the course of the disease, reaching a peak during the early-phase stages [151], possibly reflecting an initial immune response to the deposition of pathological aggregates. Furthermore, TREM2 levels have been shown to correlate with p- and t-Tau, but not with β-Amyloid 1–42 levels, suggesting that TREM2 in CSF might be a useful marker of glial activation in response to neuronal damage independent of amyloid pathology [186,187,188]. Moreover, in sporadic ALS CSF samples, the soluble TREM2 levels were higher compared to the controls [96]. Moreover, the mean levels of TREM2 were higher in the early disease phase compared to the later phases, where higher levels were associated with a slower disease progression, indicating a possible neuroprotective effect of TREM2 on microglia. Finally, in a study investigating 64 individuals with FTD, soluble TREM2 levels did not differ between FTD overall or in a particular clinical subtype of FTD; however levels were significantly higher in GRN mutation carriers as well as FTD individuals with AD-like CSF profiles (t-Tau/ Aβ 1–42 ratio > 1.0) [152].

#### 3.2.3. Progranulin

Variants in GRN are among the most frequent genetic causes of FTD, most leading to progranulin haploinsufficiency. Progranulin is a ubiquitous protein involved in several biological pathways, including growth factor-like activities, neuroinflammation, and neuronal survival [153]. Levels of progranulin can be measured in patient plasma and were shown to be decreased in both GRN mutation carriers and pre-symptomatic GRN carriers compared to controls without significant differences between the three major FTD phenotypes represented by FTD, PPA, and CBS [154]. In addition, GRN-negative FTD patients also displayed decreased levels of progranulin in CSF without correlation to CSF tau levels [189]. However, these conclusions need to be further investigated because another study analyzed progranulin concentrations in CSF of the different FTD subtypes and found levels to be significantly lower in semantic dementia and bvFTD compared to progressive non-fluent aphasia (PNFA) and controls, indicating that CSF-progranulin might represent a promising biomarker for TDP-mediated FTD instead of tau-mediated FTD [155]. In ALS, causal genetic variants in GRN, segregating with the disease, have not been observed so far, but GRN seems to act as a genetic modifier of the degree of motor neuron degeneration [190]. Importantly, levels of progranulin measured in plasma and CSF of newly diagnosed ALS patients did not differ from healthy controls, GRN-negative FTD patients, or AD [191]. Progranulin can also be found elevated in the blood of AD patients compared to controls [156].

#### 3.2.4. Synaptic Proteins

Neurogranin is a neuron-specific dendritic protein regulating synaptic plasticity and learning. Considering the central role of synaptic pathology in AD pathogenesis, neurogranin is particularly interesting as a biomarker in AD. Large studies showed that neurogranin levels in CSF specifically increased in AD compared to several other neurodegenerative diseases, including PD, FTD, ALS, and vascular dementia [157,158], confirming the potential of CSF neurogranin in separating AD dementia from non-AD dementia. On a neuropathological level, CSF neurogranin was associated with Amyloid-ß plaque pathology but not with neurofibrillary tangle of tau protein, α-synuclein, or TDP-43 loads [157]. Another synaptic protein and potentially important marker for functional synapses is synaptosomal-associated protein 25 (SNAP-25). Regarding this factor, increased levels of SNAP-25 could be detected in CSF of AD and Creutzfeldt–Jakob disease (CJD) patients but not in FTD, ALS or controls [159,160,192]. Finally, as mentioned in our previous section, NPTX2, which is involved in synaptic plasticity, was found lower in the CSF of the symptomatic phase of FTD patients regardless of the underlying genetic mutation (C9ORF72, GRN and MAPT) and, therefore, is of yet unclear significance as TDP-43 specific biomarker [95].

#### 3.2.5. Transthyretin

Another biomarker candidate is transthyretin, which is required for the transport of thyroxine and retinol but also binds with β-amyloid, and thus, is considered to prevent formation of senile plaques. Proteomic studies have demonstrated an increase in transthyretin concentration in CSF from both AD and FTD patients [99,161,162]. In addition, two studies have reported reduced levels of transthyretin in AD CSF [163,193]. These findings seem to represent an apparent contradiction but they can possibly be explained by the fact that transthyretin is modified post-translationally [194], making reliable measurement difficult [99]. Following up on this possibility, transthyretin levels were decreased in ALS CSF of one study [195], but increased in a second one [196].

#### 3.2.6. Clusterin

Clusterin is a potent chaperone that can inhibit protein aggregation, and therefore, might protect against neurotoxicity. Importantly, it has been shown that clusterin protects against TDP-43 aggregation and mislocalization [197]. In initial studies, plasma clusterin concentrations were significantly increased only in AD and not in vascular dementia, Parkinson’s disease-related dementia, dementia with Lewy bodies (DLB), or FTD [164]. One study measured clusterin in neuronal exosomes from serum and found it to be elevated in FTD, progressive supranuclear palsy (PSP) and CBS compared to α-synuclein proteinopathies [165]. In ALS, plasma levels of clusterin seem to be decreased; however, the sample sizes were small [166].

#### 3.2.7. Tau

Tau, the major microtubule associated protein of a mature neuron, is regulated by its phosphorylation state. In AD and other tauopathies, brain tau is hyperphosphorylated (p-tau) and becomes aggregated, resulting in neurofibrillary tau pathology. The CSF biomarkers for AD, particularly (p-) tau and amyloid b-42, mostly appeared to be of limited value for the diagnosis of FTD and its pathological subtypes [198,199]. However, a reduced CSF phospho-tau-181 to total tau ratio (p/t-tau ratio) has been demonstrated to be useful to discriminate FTLD-TDP from FTLD-Tau (*p* < 0.01 and *p* = 0.005) [167,200]. Intriguingly, the presence of ALS has been shown to relevantly influence the effect on this ratio [167,168].

#### 3.2.8. TDP-43

As already described, TDP-43 is the pathological protein found aggregated in neurons and glial cells of patients with ALS, FTD, and LATE. While the pathological form of TDP-43 can be detected in human post mortem brains, the detection in patient biofluids, such as CSF and serums using antibody-based assays, have been difficult [201], mainly because most antibodies bind both the pathological and physiological form of TDP-43. While most studies report elevated levels of TDP-43 in sporadic ALS independently of the genotype, an autopsy-controlled study of FTLD-TDP CSF samples reported lower levels of TDP-43 [169,170,202]. In addition, TDP-43 in the serum was also found decreased in FTD patients with the C9ORF72 mutation or FTD-MND phenotype (both subtypes are strongly associated with TDP-43 type B brain pathology) but not in FTD patients with GRN mutation [171] that are associated with Type A pathology [203]. Thus far, the detection of phosphorylated TDP-43 has failed in CSF [172,202]. In serum, however, higher phosphorylated TDP-43 levels were found in FTD C9ORF72 and GRN mutation carriers, while the same group reported lower levels of normal TDP-43 [173]. High-sensitivity assays improved the detection rate of TDP-43 in biofluids without improving the specificity to the pathological forms [204,205]. TDP-43 levels in serum were reported to be lower in AD patients compared to FTD and FTD patients with autopsy-confirmed TDP-43 pathology [174,175]. The detection of the pathological aggregated form of TDP-43 using RT-QuIC in CSF is in development and its clinical application must be further tested [59].

In summary, none of the biomarkers described above yet have the ability to differentiate between clinical phenotypes of TDP-43 pathology across the ALS/FTLD spectrum diseases. Most studies show potential biomarkers of fast progressive ALS. Only a few studies have investigated biomarkers in different FTD clinical subtypes, of which some, such as bvFTD, are more likely to have underlying TDP-43 proteinopathy. However, most studies do not compare biomarkers across all clinical cohorts of ALS, FTD, and AD sufficiently. Rather worryingly, some studies have also included clinical entities that often harbour TDP-43 pathology but without a definite proof of its pathology. In addition, definite FTD-TDP patient cohorts are usually mixed cohorts of autopsy-confirmed or genetically proven FTD-TDP and limited to small numbers. Furthermore, at the moment there have been no studies that separate confirmed LATE-NC from typical AD biosamples. This should be considered of high priority because an important development of biomarkers for AD would be to allow them to discriminate AD from definite FTD-TDP or LATE. Indeed, an initial step towards developing such a biosample cohort could be to already separate samples that lack typical AD profiles, such as TREM2, Tau, p-Tau, and Amyloid-ß changes.

## 4. Conclusions

Thus far, the development of TDP-43 as a biomarker to detect TDP-43 proteinopathies has not been successful for several reasons. In general, low concentrations in CSF and serum, as well as the detection of pathology unspecific forms are limiting such developments. Therefore, to advance TDP-43-specific biomarker development it will be necessary to further develop techniques that are aimed at detecting the pathological forms of TDP-43. As proteomic studies have yet failed to detect such forms in vivo in CSF [72,73,74], it might be necessary in the future to investigate biomarkers or techniques that are linked to its pathology, as we have described in the first part of this review. Importantly, the role of TDP-43 in regulating certain transcripts could lead to measuring changes in a range of target proteins that could lead to a differential proteome in the central nervous system and to some extent in CSF [97]. Parallel to these approaches, a still rather understudied approach could be looking at extracellular vesicles or leukocyte-derived microvesicles, which could be a potential alternative source to identify TDP-43-specific biomarkers [124,201,206]. However, while matrix effects for proteomic studies are usually lower in these species, their enrichment is often challenging. In addition, to test any pathology-derived biomarker, biosamples are needed that are well defined by TDP-43 pathology. The focus must therefore be on larger autopsy-confirmed biosample cohorts (matched biofluid tissue cohorts) or genetically confirmed TDP-43 cases. Like the amyloid-tau-neurodegeneration (ATN) classification system in AD, already established biomarkers of neurodegeneration, such as Nfs, should be routinely integrated in the search for pathology-specific biomarkers. In addition, using as a reference a combination of existing biomarkers which already have been tested in clinical cohorts could enhance the specificity of discovering novel biomarkers. The development of TDP-43-specific biomarkers is important as it may inform about TDP-43 pathology among clinical subtypes of FTD or AD cases, and could therefore enable the stratification of patients by TDP-43 pathology and eventually result in more precise treatment strategies and better interpretations of clinical trial results.

## 5. Patents

E.F. has a patent filed for a method for diagnosing a condition characterized by TDP-43 proteinopathy (PCT/GB2021/050821; https://patentscope.wipo.int/search/en/detail.jsf?docId=WO2021198698, accessed on 8. February 2023).

## Figures and Tables

**Figure 1 cells-12-00597-f001:**
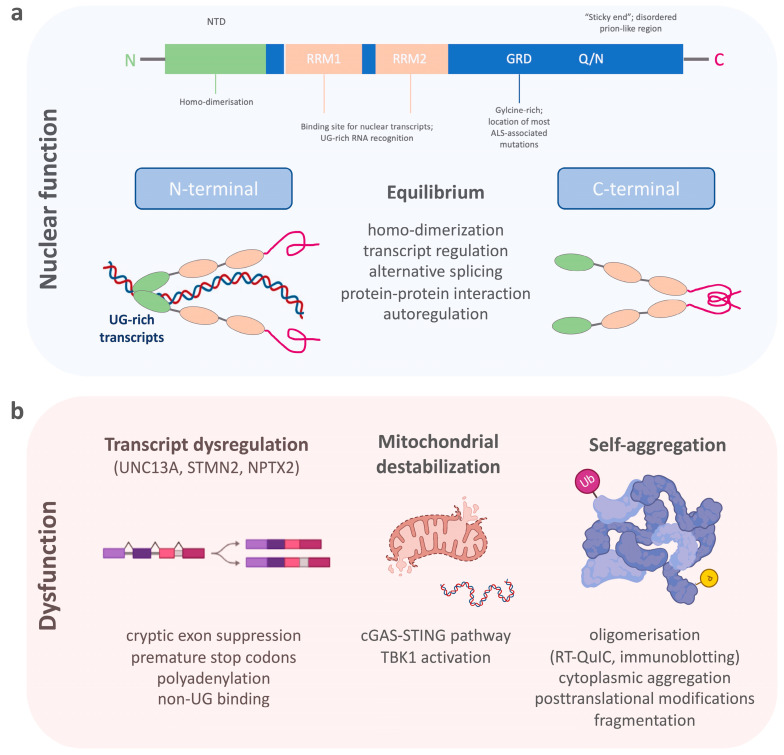
The schematic shows (**a**) the structure of TDP-43 and its importance for the physiological functions within the cell nucleus, and (**b**) the dysfunctional state of TDP-43 with the dysregulation of its nuclear function, destabilization of mitochondria, and its aggregation propensity.

**Table 1 cells-12-00597-t001:** Hypothesis driven investigation of biomarkers in ALS-FTD spectrum diseases.

Biomarker	ALS		FTD										AD	Sample	Ref.
				Clinical Subtypes				Pathology Subtypes		Genetic Subtypes					
	all	*C9ORF72*	all	bvFTD	PPA		FTD-ALS	-TDP	-Tau	*C9ORF72*	*GRN*	*MAPT*			
					SD	PNFA									
Neurofilaments ^+^	↑ (vs. controls, FTD-ALS, FTD, AD)	↑ (vs. asymptomatic *C9*-carriers, *C9*-FTD)	↑ (vs. controls, AD)	↑ (vs. controls)	↑ (vs. controls)	↑ (vs. controls)	↑ (vs. controls, bvFTD, PPA) – (vs. FTD)	Not sufficiently studied (*n* = 2)	Not sufficiently studied (*n* = 2)	↑ (vs. asympomatic C9-carriers, sporadic FTD)	↑ (vs. asympomatic *GRN*-carriers, sporadic FTD, *C9*-FTD, *MAPT*-FTD)	↑ (vs. asympomatic *MAPT*-carriers)	↑ (vs. controls)	CSF Serum	[144,145,146,147,148,149,150]
CHIT1	↑ (vs. controls, FTD, AD)	↑ (vs. controls, asymptomatic carriers, genetic FTD) *	↑ (vs. controls, not vs. ALS, AD)	– (vs. PPA, FTD-ALS)	– (vs. bvFTD, FTD-ALS)		– (vs. bvFTD, PPA)	Not sufficiently studied	Not sufficiently studied	↑ (vs. controls, asymptomatic carriers) *	Not studied	Not studied	↑ (vs. controls, not vs. FTD)	CSF	[109,112,113]
YKL-40	↑ (vs. controls, not vs. ALS)	↑ (vs. controls, asymptomatic carriers, not vs. genetic FTD) *	↑ (vs. controls, slighty vs. ALS, not vs. AD)	– (vs. PPA, FTD-ALS)	– (vs. bvFTD, FTD-ALS)		↑ (vs. bvFTD, not PPA)	Not sufficiently studied	Not sufficiently studied	↑ (vs. controls, asymptomatic carriers, not vs. genetic ALS) *	Not studied	Not studied	↑ (vs. controls, not vs. FTD)	CSF	[112,113]
GFAP	– (vs. controls)	– (vs. controls, asymptomatic carriers) *	↑ (vs. controls, ALS, not vs. AD)	– (vs. PPA, FTD-ALS)	– (vs. bvFTD, FTD-ALS)		– (vs. bvFTD, not PPA)	Not sufficiently studied	Not sufficiently studied	↑ (vs. controls, asymptomatic carriers, genetic ALS) *	Not studied	Not studied	↑ (vs. controls, not vs. FTD)	CSF	[112,113]
TREM2	↑ (vs. controls)	*Not studied*	– (vs. controls)	– (vs. controls, PPA)	– (vs. controls, bvFTD)	– (vs. controls, bvFTD)	Not studied	Not studied	Not studied	– (vs. controls, *C9*-FTD, *MAPT*-FTD)	↑ (vs. controls, *C9*-FTD, *MAPT*-FTD)	– (vs. controls, *C9*-FTD, *MAPT*-FTD)	↑ (vs. controls)	CSF	[96,151,152]
Progranulin	– (vs. controls, *GRN*-negative FTD, AD)	*Not studied*	↓ (vs. controls)	↓ (bvFTD and SD vs. controls, PNFA)	↓ (SD and bvFTD vs. controls, PNFA)	– (vs. controls)	Not studied	Not studied	Not studied	– (vs. controls)	↓ (vs. controls) – (vs. asymptomatic carriers)	Not studied	↓ (vs. controls)	Plasma CSF	[153,154,155,156]
Neurogranin	↓ (vs. AD)	*Not studied*	↓ (vs. AD, controls)	↓ (vs. AD) – (vs. SD, PNFA)	↓ (vs. AD) – (vs. bvFTD, PNFA)	↓ (vs. AD) – (vs. bvFTD, SD)	Not studied	Not studied	Not studied	Not studied	Not studied	Not studied	↑ (vs. controls, ALS, FTD)	CSF	[157,158]
SNAP-25	Not studied	Not studied	– (vs. controls)	Not studied	Not studied	Not studied	Not studied	Not studied	Not studied	Not studied	Not studied	Not studied	↑ (vs. controls)	CSF	[159,160]
Transthyretin	↑/↓ (vs. controls)	Not studied	↑ (vs. controls)	Not studied	Not studied	Not studied	Not studied	Not studied	Not studied	Not studied	Not studied	Not studied	↑/↓ (vs. controls)	CSF	[161,162,163]
Clusterin	↓ (vs. controls)	Not studied	↑ (vs. controls)	Not studied	Not studied	Not studied	Not studied	Not studied	Not studied	Not studied	Not studied	Not studied	↑ (vs. controls)	Serum plasma	[164,165,166]
p/t ratio	Not studied	Not studied	↓ (vs. controls)	↓ (vs. controls)	↓ (vs. controls)	↓ (vs. controls)	↓ (vs. controls, FTD without ALS, AD)	Not sufficiently studied	Not sufficiently studied	Not studied	Not studied	Not studied	↓ (vs. controls, FTD)	CSF	[150,167,168]
TDP-43	↑ (vs. controls, FTD)	↑ (vs. controls, FTD, *C9*-FTD)	↓ (vs. ALS, *C9*-ALS) ↑ (vs. controls)	↓ (vs. controls)	– (vs. controls, FTD)	– (vs. controls, FTD)	*↓* (vs. FTD, controls)	↓ (vs. FTD-Tau)	Not sufficiently studied	↓ (vs. *GRN*-FTD)	↑ (vs. *C9*-FTD)	↑ (vs. FTD-TDP)	↓ (vs. FTD-TDP, FTD, AD-TDP) ↑ (vs. controls)	CSF serum	[169,170,171,172,173,174,175]

↑/↓ arrows indicate increased or decreased biomarker levels. Arrows do not always reflect statistical significance. + includes neurofilament light and phosphorylated heavy chain levels, * genetic ALS/FTD with the majority being C9ORF72-carriers. AD—Alzheimer’s disease, ALS—Amyotrophic lateral sclerosis, FTD—Frontotemporal dementia, bvFTD—Behavioral variant frontotemporal dementia, GRN—Granulin, PNFA—Progressive non-fluent aphasia, PPA—Primary progressive aphasia, SD—Semantic dementia.

## Data Availability

The data presented in this study are openly available in [Pubmed].

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
