# Peer review of "TDP-43 Proteinopathy Specific Biomarker Development"

_cells, 2023, doi:10.3390/cells12040597_

Round 1

Reviewer 1 Report

TDP-43 is a primary or secondary trademark in many neurodegenerative diseases.  This review discusses in great detail TDP-43 specific biomarkers that might be useful in categorizing neurodegenerative diseases based on the underlying pathophysiology related to TDP-43 proteinopathy as opposed to categorizing clinical subgroups of neurodegenerative disease. This review demonstrates how this focus could lead to specific therapies to target the underlying pathophysiology (precision medicine).

The review is very well written and is very thorough, covering all the pertinent literature related to this topic. I wholeheartedly endorse accepting the manuscript.

Author Response

Thank you for the positive response on our article. Reviewer 1 does not ask for revisions to be adressed. 

Reviewer 2 Report

The review paper entitled ‘TDP-43 proteinopathy specific biomarker development’ summarizes and comments on the most recent literature describing biomarkers linked to TDP-43 pathology. The authors discuss biomarkers linked to pathophysiological processes involved in the development of TDP-43 proteinopathy and studies analyzing biofluids from clinical cases associated to TDP-43 pathology. The review is very well-written, insightful, and clearly structured. I think this review is an excellent addition to the literature providing a ‘to-do list’ of what is missing, and it needs to be achieved by the field. I don’t have any specific requests. I would only suggest:

1-     Consider PPIA as another potential biomarker because of a recent publication showing that loss of PPIA induces TDP-43 pathology in mice and it is linked to a FTD behavior. Furthermore, lower levels of PPIA correlated with a faster progression of ALS.

2-     Reporting that phosphorylated TDP-43 was observed outside of extracellular vesicles in an immunogold experiment performed on blood-derived extracellular vesicles. This could be added as a warning in the analysis of TDP-43 associated with EVs (lines 564-565). 

Author Response

Dear Reviewer 2, please find our point by point answers below,

1-     Consider PPIA as another potential biomarker because of a recent publication showing that loss of PPIA induces TDP-43 pathology in mice and it is linked to a FTD behavior. Furthermore, lower levels of PPIA correlated with a faster progression of ALS.

We added this information to our text in section 2.5. Molecular aspects of TDP-43 pathology:

“For example, a loss of function mutation in cyclophilin A has been found in ALS and knocking out cyclophilin A in a mouse model led to the development of TDP-43 proteinopathy [128]. Cyclophilin A has also been found differentially expressed in ALS peripheral blood monocytes and plasma-derived extracellular vesicles [129,130].” (line 340-443)

2-     Reporting that phosphorylated TDP-43 was observed outside of extracellular vesicles in an immunogold experiment performed on blood-derived extracellular vesicles. This could be added as a warning in the analysis of TDP-43 associated with EVs (lines 564-565).

We added the reference {130} of this finding to our article (line 624)
